# Cardiac Vagal Control, Regulatory Processes and Depressive Symptoms: Re-Investigating the Moderating Role of Sleep Quality

**DOI:** 10.3390/ijerph16214067

**Published:** 2019-10-23

**Authors:** Sarah K. Danböck, Gabriela G. Werner

**Affiliations:** 1Clinical Stress and Emotion Laboratory, Division of Clinical Psychology and Psychopathology, Department of Psychology, University of Salzburg, Hellbrunner Str. 34, 5020 Salzburg, Austria; SarahKatharina.Danboeck@sbg.ac.at; 2Division of Clinical Psychology and Psychological Treatment, Department of Psychology, Ludwig-Maximilians-University Munich, Leopoldstr. 13, 80802 Munich, Germany

**Keywords:** cardiac vagal control, regulatory processes, sleep quality, depressive symptoms

## Abstract

Lower cardiac vagal control (CVC), which is often understood as an indicator for impaired regulatory processes, is assumed to predict the development of depressive symptoms. As this link has not been consistently demonstrated, sleep quality has been proposed as a moderating factor. However, previous studies were limited by non-representative samples, cross-sectional data, and focused on CVC as a physiological indicator for impaired regulatory processes, but neglected corresponding subjective measures. Therefore, we investigated whether sleep quality moderates the effects of CVC (quantified by high-frequency heart rate variability) and self-reported regulatory processes (self- and emotion-regulation) on concurrent depressive symptoms and on depressive symptoms after three months in a representative sample (*N* = 125). Significant interactions between CVC and sleep quality (in women only), as well as self-/emotion-regulation and sleep quality emerged, whereby higher sleep quality attenuated the relation between all risk factors and current depressive symptoms (cross-sectional data). However, there were no significant interactions between those variables in predicting depressive symptoms three months later (longitudinal data). Our cross-sectional findings extend previous findings on sleep quality as a protective factor against depressive symptoms in the presence of lower CVC and subjective indices of impaired regulatory processes. In contrast, our conflicting longitudinal results stress the need for further investigations.

## 1. Introduction

### 1.1. Cardiac Vagal Control and Regulatory Processes

Cardiac vagal control (CVC), which is indexed by the variation in heart rate related to respiration (respiratory sinus arrythmia), is associated with the functioning of the parasympathetic nervous system, which is assumed to have broad homeostatic functions [1,2]. CVC itself has been proposed as a proxy for the inhibitory capacity of the central autonomic network, which regulates behavioral, cognitive, and emotional responses [3,4]. In line with this, higher CVC has been associated with the use of adaptive emotion regulation strategies, better down-regulation of negative affect, and more flexible emotional responding, and is therefore assumed to be a physiological marker of emotion regulation [5,6]. Moreover, higher CVC has also been linked to other facets of self-regulation, like attention control and cognitive control [6].

### 1.2. Lower CVC as a Risk Factor for Depressive Symptoms

Lower CVC has often been cross-sectionally associated with diagnosis and symptoms of depression [7]. More precisely, lower levels of resting CVC have been linked to higher depressive symptoms in participants with clinical depression [8], remitted depression [9], healthy but high-risk population groups [10], and healthy participants [11]. Recent longitudinal studies support the assumption that lower CVC also predicts the development of depressive symptoms [12,13,14]. For instance, lower resting CVC at baseline assessment predicted depressive symptoms after one year in healthy adolescents [12] and healthy young adults [13]. In line with this, experimentally induced increases in CVC reduced depressive symptoms in participants who had undergone cardiac surgery [15] and in participants with clinical depression [16], suggesting a causal role of CVC in the development of depression.

Altogether, most previous findings support a relationship between lower CVC and greater depressive symptoms. Moreover, first longitudinal and experimental studies point toward CVC as a precursor of depressive symptoms. However, there are also some inconsistent cross-sectional and longitudinal findings showing positive links or no links between CVC and depressive symptoms [17,18,19], which implies the need to investigate potential moderators of the relation between CVC and depressive symptoms.

### 1.3. Sleep Quality as a Protective Factor Against the Development of Depressive Symptoms in the Presence of Other Risk Factors

Whereas people with severe sleep disturbances have a twofold risk of developing depression [20,21], good sleep quality might facilitate self-regulatory as well as neurophysiological processes [22] and serve as a protective factor against the development of depression. Therefore, high sleep quality has been proposed as a protective factor against the development of depressive symptoms in the presence of other risk factors [23,24]: when risk factors like lower CVC or impaired regulatory processes increase the risk for depressive symptoms, high sleep quality could offset the risk by providing additional regulatory resources. In other words, risk factors and sleep quality might interact, such that the risk factors only increase depressive symptoms when sleep quality is also low.

To date, only two studies have directly tested sleep quality as a moderator of the relation between CVC and depression. First, El-Sheik, Erath and Keller [25] assessed sleep, depressive symptoms, and CVC in a large sample (*n* = 167) of healthy boys and girls aged between eight and nine years. In line with the proposed model, CVC and sleep quality interacted in predicting depressive symptoms, whereby low levels of CVC were only linked to greater depressive symptoms when sleep quality was low [25]. Second, Werner et al. [23] replicated these results in a small sample (*n* = 29) of healthy young women aged between 19 and 31 years. Altogether, both studies support the idea of sleep as a protective factor against depressive symptoms in the presence of a lower CVC. However, both studies are based on very homogenous samples, either children [25] or women [23], which restricts the generalizability of findings. Due to the sole use of cross-sectional data, the direction and causal nature of these relations also remain unclear. To date, no study has investigated whether sleep quality also functions as a protective factor against the development of depressive symptoms in the presence of impaired regulatory processes, which are supposed to be indexed by lower CVC [3,4]. Furthermore, other, more direct measures of these regulatory processes have not yet been investigated. As impaired self- and emotion-regulation processes themselves have also been suggested to increase the risk of developing depressive symptoms [26,27,28], high sleep quality might serve as a protective factor in this context as well.

### 1.4. The Present Study

The present study was designed to replicate and extend previous findings on sleep quality as a protective factor against the development of depressive symptoms in the presence of lower CVC or other measures of impaired regulatory processes. Using a combined cross-sectional and longitudinal design and a large, representative sample of healthy adults (including women and men), we specifically aimed to address concerns regarding the interpretability and generalizability of earlier studies [23,25]. As CVC might be a better index for regulatory processes in women than in men [29], influences of sex on the proposed moderation model were also investigated for exploratory purposes.

We first assessed trait-level resting CVC, trait-level subjective self- and emotion-regulation, subjective sleep quality, and concurrent depressive symptoms, and then assessed depressive symptoms again three months later. Specifically, we assessed resting CVC by quantifying heart rate variability in the high-frequency spectral band (HF-HRV; 1) during an extended resting baseline while participants watched a neutral film following [23]. Self- and emotion-regulation, sleep quality, and depressive symptoms were assessed using established questionnaires. Just like Werner et al. [23], we expected sleep quality to interact with CVC in predicting same-time and later depressive symptoms. More precisely, we expected lower CVC to predict higher depressive symptoms only when sleep quality was also low. In addition, we expected sleep quality to interact with self- and emotion-regulation in predicting same-time and later depressive symptoms in the same manner.

## 2. Materials and Methods

### 2.1. Participants

Only healthy adults who had no self-reported psychiatric, cardiovascular, neurological, or thyroid diseases could participate in the study (overall inclusion criteria). Participants who smoked on a regular basis (≥10 cigarettes a day) or who took medicine on the day of assessment that might have interacted with the (para-) sympathetic nervous system (antihistamines, painkillers, sympathomimetics, cortisone and sumatriptan) were excluded from analyses including HF-HRV; participants with missing or irregular HF-HRV data were excluded from those analyses as well (HF-HRV inclusion criteria).

A sample-size calculation (two-sided, power = 0.80, alpha = 0.05; G*Power3.1) based on the observed R^2^ increase (0.11) when the interaction between HF-HRV during a neutral film and sleep quality was added to the regression model used in the study by Werner et al. [23] indicated a sample size of *n* = 66 as the minimum necessary to detect a statistically significant interaction between HF-HRV and sleep quality in predicting depressive symptoms. To compensate for potential non-fulfillment of inclusion criteria and dropouts, we initially recruited 140 participants. Of those, 125 participants (69.6% women) aged between 18 and 35 years met the overall inclusion criteria (for descriptive values see Table 1) and 112 participants (68.8% women) were included in HF-HRV analyses. Due to dropouts, longitudinal data was available for 84 participants (72.6% women) who met overall inclusion criteria and 75 participants (70.7% women) who met HF-HRV inclusion criteria. Participants who dropped out did not differ from completers in CVC, self-regulation, emotion regulation, sleep quality, and depressive symptoms, *p*s ≥ 0.613.

### 2.2. Procedure

The study consisted of two sessions at intervals of 3 months. Session 1 (T1) took part in the clinical psychology lab of the Ludwig-Maximilians-University Munich (duration: 60 min). After inclusion criteria were checked, participants were seated on a chair placed 50 cm away from an 18-inch computer monitor, and electrodes for measuring cardiovascular activity were attached. Cardiovascular activity was recorded while participants were asked to look at a fixation cross (2-min baseline) and then viewed a neutral film with nature scenes (10-min extended baseline). Stimulus presentation was controlled by E-Prime 2.0 (Psychology Software Tools, Inc., Pittsburgh, PA, USA). Afterwards, self-regulation, emotion regulation, sleep quality, depressive symptoms, and sociodemographic information were assessed using established questionnaires. Moreover, body mass index (BMI) was computed using participants’ self-reported weight and height. After three months, Session 2 (T2) was conducted online (duration: 10 min), and self-regulation, emotion regulation, sleep quality, and depressive symptoms were once again assessed. Yet, only the re-assessment of depressive symptoms was used for the current analyses.

The study was approved by the local ethics committee. Participants signed written informed consent before the study and were compensated with course credit or payment of €8 for attending Session 1. They were included in a voucher lottery if they also took part in Session 2.

### 2.3. Measures

#### 2.3.1. CVC

Given that the current study was a replication and an extension of Werner et al. [23], we assessed and preprocessed CVC in the same way: we assessed resting CVC by quantifying HF-HRV for a 10-min extended baseline during which participants watched a neutral film displaying nature scenes accompanied by instrumental music. Psychophysiological measurements during the pre-film baseline and the neutral film were recorded using the amplifier Biopac Systems MP150 and the recording software package AcqKnowledge 4.0 with a sampling rate of 1000 Hz. The recordings contained electrocardiogram (ECG), for which alcohol pads were used to clean the skin sites. ECG was recorded using disposable 35-mm diameter solid-gel snap electrodes; the electrodes were applied on the upper sternum and lowest rib on the left side. A 0.05 Hz highpass filter was applied during ECG measurement. After recording, ECG raw data were bandpass filtered between 0.5 and 40 Hz and further processed using the software Autonomic Nervous System Laboratory (ANSLAB 2.6; University of Salzburg, Salzburg, Austria) [30,31]; R-spikes were determined automatically by ANSLAB and checked further manually.

In ANSLAB, the preprocessed ECG was analyzed with power spectral analyses between 0.15 Hz and 0.40 Hz. More specifically, to obtain HF-HRV values, heart-period time series were linearly detrended and resampled at 4 Hz using cubic spline interpolation. Then, power-spectral densities were computed using the Welch algorithm, which creates ensemble averages of successive periodograms. The averages were derived from spectra estimated for 120-s segments, overlapping by half. We used Hanning-windowed segments that were subjected to fast Fourier transform. Estimates of power were adjusted to account for attenuation produced by the Hanning window, and distribution characteristics were normalized by natural-logarithm transformation.

Although it has been suggested that associations between cardiac vagal activity and HF-HRV might be influenced by aspects of respiration [2], we did not adjust for respiration, because people might differ in respiratory function due to factors unrelated to CVC, such as basal metabolic rate and respiratory pacemaker function [2]. However, we did control for other possible confounding factors, like age and BMI following [23], and included interactions with sex in the HF-HRV analyses as well.

#### 2.3.2. Self-Regulation

To assess a broad range of self-regulatory processes, the Self-Regulation Scale (REG) [32] was used. This 10-item self-report scale measures the ability to maintain focus on a task when distractors are present or setbacks have occurred. Therefore, the REG is thought to cover both aspects of attention regulation (e.g., “I can control my thoughts from distracting me from the task at hand”) and emotion regulation (e.g., “If an activity arouses my feelings too much, I can calm myself down so that I can continue with the activity soon“). Participants rate whether the statements apply to them on a 4-point scale from 1 (not at all true) to 4 (exactly true). Items were recoded so that higher (sum) scores indicate better self-regulation (range: 10–40). The REG showed acceptable internal consistency (T1: *α* = 0.79) in this sample.

#### 2.3.3. Emotion Regulation

To assess emotion regulation, the Difficulties in Emotion Regulation Scale (DERS) [33,34] was used. The DERS is a 36-item self-report measure assessing individuals’ regular levels of emotion dysregulation across six domains: nonacceptance of negative emotions, inability to engage in goal-directed behaviors when distressed, difficulty controlling impulsive behaviors when distressed, limited access to emotion regulation strategies perceived as effective, lack of emotional awareness, and lack of emotional clarity. Responders indicate how often the given statements apply to them on a 5-point scale from 1 (almost never) to 5 (almost always). Items were recoded so that higher scores indicate greater difficulty in emotion regulation with a sum score of 36 to 180. In the present sample, internal consistency for the DERS was excellent (T1: *α* = 0.91).

#### 2.3.4. Sleep Quality

To assess overall subjective sleep quality, the Pittsburgh Sleep Quality Index (PSQI) [35,36] was used. The PSQI assesses self-reported sleep quality and sleep disturbances over the preceding four weeks. Eighteen items generate seven component scores (range: 0–3): subjective sleep quality, sleep latency, sleep duration, habitual sleep efficiency, sleep disturbances, use of sleeping medication, and daytime dysfunction. These scores are summed up to obtain one global measure of overall subjective sleep quality (range: 0–21). Lower values in the PSQI indicate better subjective sleep quality. A PSQI score greater than 5 indicates severe difficulties in at least two areas or moderate difficulties in more than three areas and is therefore suggested as the clinical cutoff. The internal consistency of the PSQI in the current sample was questionable (T1: *α* = 0.59).

#### 2.3.5. Depressive Symptoms

Depressive symptoms were measured using the Beck Depression Inventory-II (BDI-II) [37,38]. This 21-item self-report questionnaire assesses the presence and severity of depressive symptoms during the preceding two weeks. For each item, participants could choose between four statements describing different degrees of symptom severity (range: 0–3). Higher scores index higher depression severity. All items were summed up (range: 0–63), whereby BDI-II scores higher than 14 indicate mild depressive symptoms. For analytic purposes, an adjusted sum score without items referring to sleep (items 16 and 20) was used to decrease overlap with sleep quality (BDI-II’; range: 0–57). The BDI-II demonstrated good internal consistency (T1: *α* = 0.87; T2: *α* = 0.90).

### 2.4. Statistical Analyses

Because most variables were non-normally distributed (Kolmogorov–Smirnov < 0.05), we used Spearman correlations to investigate relationships between CVC, self-regulation, emotion regulation, sleep quality, current depressive symptoms and depressive symptoms after three months. Additionally, on the basis of current recommendations [39], links with further confounding variables (i.e., age, BMI) were examined. To explore sex differences in primary variables, *t*-tests were conducted.

To investigate whether sleep quality serves as a protective factor against depressive symptoms in the presence of other risk factors, like lower CVC or impaired regulatory processes, we conducted moderation analyses using the PROCESS tool by Hayes (version 3.3) [40]. Following the recommendations of Long and Ervin [41], as well as Hayes and Cai [42], heteroscedasticity-consistent standard error estimators (HC3) were used to make sure that conclusions were not compromised by heteroscedasticity, which was present in our current data. Continuous independent variables were mean-centered (i.e., CVC, self-regulation, emotion regulation, sleep quality), and dichotomous independent variables were dummy coded (i.e., sex; 0 = female, 1 = male). Dependent variables (i.e., depression at T1, depression at T2) and covariates (i.e., depression at T1) remained unchanged. For (marginally) significant interactions, conditional effects at plus and minus one standard deviation from the moderator mean were reported (simple slope analyses). All moderation analyses were conducted twice: first, based on cross-sectional data (outcome: depressive symptoms at T1) and, second, on longitudinal data (outcome: depressive symptoms at T2). In all longitudinal analyses, depressive symptoms at T1 were included as a covariate to control for baseline differences in depressive symptoms. We conducted moderation analyses with CVC, sleep quality, and the interaction between CVC and sleep quality as predictors of depressive symptoms at T1 and T2. As first evidence suggests that CVC is a better indicator for regulatory capacities in women than in men, additional models with sex as a second moderator were also calculated in exploratory analyses. Furthermore, we generated moderation models with self-regulation, sleep quality, and the interaction term between self-regulation and sleep quality as predictors for depressive symptoms at T1 and T2. Finally, similar regression models were calculated for emotion regulation instead of self-regulation.

## 3. Results

Descriptive statistics and zero-order correlations among CVC, self-regulation, emotion regulation, sleep quality, and depressive symptoms, as well as possible confounding variables, are summarized in Table 1. The means of depressive symptoms and sleep quality were in the normal range. Ten participants (8%) exceeded the clinical cutoff (>14) of the depression scale at T1 and nine participants (11%) at T2, indicating at least mild depressive symptoms. Thirty-seven participants (30%) revealed values above the suggested cutoff (>5) for poor sleep at T1.

All primary variables significantly correlated with each other in the expected direction except for CVC. As expected, higher CVC was significantly linked to fewer difficulties in emotion regulation. However, it was only marginally linked to higher self-regulation and better sleep quality, and, unexpectedly, it was not associated with depressive symptoms at T1 and T2. None of the possible confounding variables were significantly related to any primary variable. T-Tests revealed no sex differences in all primary variables (*p*s ≥ 0.258), yet results showed marginally significant higher CVC for women (*M* = 7.50, *SD* = 1.24) than men (*M* = 7.06, *SD* = 1.24), *t* (110) = 1.75, *p* = 0.084.

### 3.1. Sleep Quality as Moderator in the Relationship between CVC and Depressive Symptoms

To test the hypothesis that subjective sleep quality acts as a moderator in the relationship between CVC and current, as well as later depressive symptoms, we conducted moderation analyses (see Table 2).

In a first cross-sectional model including CVC, sleep quality, and their interaction as predictors (model C1), no significant interaction effect emerged. However, when sex was added as a second moderator (model C2), a marginally significant three-way interaction between sex, CVC, and sleep quality was observed. Whereas CVC and sleep quality interacted in predicting concurrent depressive symptoms in women (*b* = −0.62, *F* (1,104) = 6.49, *p* = 0.012), they did not interact in predicting depression in men (*b* = 0.11, *F* (1, 104) = 0.15, *p* = 0.703). Further, simple slope analyses revealed that for women with higher sleep quality (lower PSQI scores), lower CVC did not predict depressive symptoms, *b* =0.35, 95% CI [−0.80, 1.49], *t* =0.60, *p* = 0.552. Yet for women with lower sleep quality (higher PSQI scores), lower CVC significantly predicted higher depressive symptoms, *b* = −2.36, 95% CI [−4.03, −0.69], *t* = −2.80, *p* = 0.006 (see Figure 1a). For men, lower CVC did not predict higher depressive symptoms, neither for men with higher sleep quality (lower PSQI scores), *b* = 0.10, 95% CI [−1.42, 1.61], *t* = 0.13, *p* = 0.898, nor for men with lower sleep quality (higher PSQI scores), *b* = 0.59, 95% CI [−1.16, 2.34], *t* = 0.67, *p* = 0.503 (see Figure 1b).

However, in the longitudinal models, CVC and sleep quality did not interact in predicting depressive symptoms three months later. This was the case in both a simple model including only CVC, sleep quality, and their interaction as predictors (including baseline depressive symptoms as a covariate; model L1), and also in a more complex model with sex and the corresponding interactions as additional predictors (model L2). The pattern of all results (in models C1, C2, L1, and L2) remained unchanged when sex, age, and BMI (models C1 and L1) and age and BMI respectively (models C2 and L2) were included as covariates.

### 3.2. Sleep Quality as Moderator in the Relationship between Impaired Regulatory Processes and Depressive Symptoms

To test the hypothesis that subjective sleep quality acts as a moderator in the relationship between self-reported measures for regulatory processes, particularly self- and emotion-regulation, and concurrent, as well as later depressive symptoms, we conducted further separate moderation analyses (see Table 3).

Self-regulatory abilities and sleep quality significantly interacted in predicting concurrent depressive symptoms (model C3). Subsequent simple slope analyses showed that for people with low sleep quality, poor self-regulation was linked to more depressive symptoms (*b* = −1.02, *SE* = 0.20, 95% CI [−1.41, −0.62], *t* = −5.10, *p* < 0.001). Yet, in people with high sleep quality, self-regulation did not influence depressive symptoms (*b* = −0.32, S*E* = 0.16, 95% CI [−0.64, 0.00], *t* = −1.97, *p* = 0.051) (see Figure 2a).

In line with this, difficulties in emotion regulation and sleep quality also interacted in predicting concurrent depressive symptoms (model C4). Simple slope analyses showed that for people with both lower and higher sleep quality, greater difficulty in emotion regulation significantly predicted more current depressive symptoms. However, this effect was much more pronounced in people with low sleep quality (*b* = 0.26, *SE* = 0.04, 95% CI [0.19, 0.33], *t* = 6.99, *p* < 0.001) than in people with high sleep quality (*b* = 0.13, *SE* = 0.04, 95% CI [0.04, 0.21], *t* =2.99, *p* = 0.003) (see Figure 2b). Contrary to these findings, neither self-regulation nor emotion regulation abilities interacted with sleep quality in predicting depressive symptoms three months later (models L3 and L4). In all longitudinal models, baseline depressive symptoms were included as a covariate. The pattern of results in models C3, C4, L3, and L4 remained unchanged when sex, age, and BMI were included as covariates.

## 4. Discussion

Sleep quality has recently been proposed as a protective factor against the development of depressive symptoms in the presence of lower CVC, which is assumed to increase the risk for depression. As very first cross-sectional findings have supported this assumption [23,25], the present study aimed to replicate and extend these preliminary results in multiple ways.

First, we tried to replicate the finding that sleep quality functions as a moderator in the relationship between CVC and concurrent depressive symptoms in a larger adult sample including both men and women. A three-way interaction between sex, sleep quality, and CVC supported this model in women but not in men. Consistent with Werner et al. [23], only women with lower sleep quality and lower CVC showed elevated depressive symptoms, whereas this was not the case in women with higher sleep quality. For men, no such protective role of sleep quality in the presence of lower CVC emerged; associations between CVC and depressive symptoms were not found either for men with higher sleep quality or for men with lower sleep quality. Thus, our findings confirm the idea that higher sleep quality can offset the risk for higher depressive symptoms in the context of lower CVC, at least in women. We assessed sleep quality during the preceding four weeks as a more state-like variable, whereas CVC during an extended baseline is considered reasonably stable in time [43] and thus conceptualized as a more trait-like variable. Our interpretation, therefore, is that high sleep quality during a certain time might counteract concurrent depressive symptoms in the context of a more stable risk factor like CVC. Nevertheless, it is also plausible that generally high CVC counteracts depressive symptoms in phases of poor sleep quality. One reason for the non-existent interaction effect in men might be a weaker relationship between CVC and depressive symptoms in men than in women. Although no prior studies have taken sex differences in the relationship between CVC and depressive symptoms into account, a recent study denotes that CVC is a better index for regulatory processes in women than in men [29]. Furthermore, as it is assumed that the link between CVC and depressive symptoms is explained by its function as an indicator for regulatory processes [23], it is also plausible that the link between CVC—as an indicator of regulatory processes—and depressive symptoms might be more pronounced in women than in men.

Second, we wanted to extend previous cross-sectional findings by also using subjective measures for regulatory processes, like self- and emotion-regulation, which are broadly assumed to be indexed by CVC [3,4,5,6]. In line with our expectations, sleep quality moderated the relationships between self- and emotion-regulation and depressive symptoms such that participants with lower sleep quality showed much stronger (about double-sized) relationships between regulatory abilities and depressive symptoms compared to participants with higher sleep quality. Therefore, good sleep quality might be able to weaken the relationship between regulatory abilities and depressive symptoms. This might indicate that when daytime habitual self- and emotion-regulation does not work well, sleep as a night-time regulatory mechanism can provide additional resources to attenuate current depressive symptoms. Conversely, good daytime habitual self- and emotion-regulation could as well attenuate depressive symptoms when sleep quality is poor. As these effects were not restricted only to women, the findings stress the idea that sleep quality serves as a protective factor against depressive symptoms in both women and men, and that our results using CVC as an indicator for impaired regulatory processes might be compromised by a weaker influence of CVC on depressive symptoms in men than in women.

Third, we re-assessed depressive symptoms after three months to test our hypothesis about the temporal relations between variables. Previous longitudinal studies indicated that sleep disturbances [21], lower CVC [12,13,14], and impaired regulatory processes [26] precede the development of depressive symptoms. Therefore, we assumed that baseline sleep quality might attenuate effects of baseline CVC, self-, and emotion-regulation on depressive symptoms at follow-up. However, contrary to our expectations, baseline sleep quality did not moderate the associations between baseline CVC and later depressive symptoms (neither in a simple model nor when sex was included as a second moderator), or between baseline self-/emotion-regulation and later depressive symptoms. The results also showed no effects of the single predictors (i.e., CVC, self-regulation, emotion regulation) on depressive symptoms at follow-up; the only factor predicting depressive symptoms at follow-up was baseline depressive symptoms. This is in contrast to previous longitudinal studies [12,13,14,21,26] and might be due to the short time interval between baseline assessment and follow-up in the present study (three months), which strongly differs from those used in previous studies (at least one year). It is possible that the influence of the mentioned risk factors may just take longer than three months to lead to manifest (changes in) symptoms of depression. Other previous studies support this assumption, as they also did not find prospective associations between difficulties in emotion regulation and depressive symptoms after shorter time intervals of, for example, two weeks [44] or seven months [45]. Thus, it is plausible that an existing interaction effect might just take longer than three months to emerge and was therefore not yet detectable in the current study. However, it is also possible that sleep quality does not serve as a protective factor against the development of future depressive symptoms and that our cross-sectional findings solely indicate that lower sleep quality, lower CVC, and high depressive symptoms are co-occurring symptoms that share a common origin and therefore specifically emerge in combination. Altogether, the present study does not allow differentiating whether the lack of longitudinal interaction effects is due to methodological shortcomings or whether it indicates that high sleep quality does not protect people from developing future depressive symptoms when regulatory processes are impaired. To clarify this issue, future studies with longer time-intervals between baseline and follow-up assessment are needed.

Some further limitations need to be considered when interpreting the current findings. First, as just mentioned, the time frame between baseline and follow-up assessment might have been too short for longitudinal effects to emerge. Without other longitudinal studies with longer follow-ups (>one year), reliable conclusions about temporal associations in our model cannot be drawn. Second, some participants dropped out before the follow-up, which reduced the power for our longitudinal analyses. However, as our a priori sample-size calculation indicated a need for at least 66 participants, we feel confident that our longitudinal analyses (based on at least 75 participants) were not underpowered. Third, correlations between CVC and self-reported regulatory processes were partly non-significant, which conflicts with the broadly accepted assumption that CVC serves as an objective indicator for self- and emotion-regulation abilities [5,6]. However, as mentioned earlier, this is in line with recent findings indicating that CVC might serve as a better indicator for regulatory processes in women than in men [29]. This is also reflected by higher correlations between CVC and self-reported regulatory processes in women (difficulties in emotion regulation: *r* = −0.23; *p* = 0.041; self-regulation abilities: *r* = 0.22; *p* = 0.055) than in men (difficulties in emotion regulation: *r* = −0.14; *p* = 0.408; self-regulation abilities: *r* = 0.06; *p* = 0.740) in the current sample. Fourth, the calculation of the BMI was based on self-reported height and weight, which might differ from objective assessments [46]. Fifth, the PSQI, which we used to assess sleep quality, showed low internal consistency.

## 5. Conclusions

The present study replicates and extends previous findings on sleep quality as a moderator of the relationship between impaired regulatory processes indexed by lower CVC and depressive symptoms. It is the first study in the field to take sex differences into account: the association between lower CVC and concurrent depressive symptoms, as well as the moderating role of sleep quality, only emerged in women and not in men. Furthermore, results demonstrated a protective function of high sleep quality against concurrent depressive symptoms in the presence of other indicators of impaired regulatory processes (i.e., self-/emotion-regulation). However, there were no results regarding depressive symptoms after three months. Therefore, further prospective investigations are needed.

## Figures and Tables

**Figure 1 ijerph-16-04067-f001:**
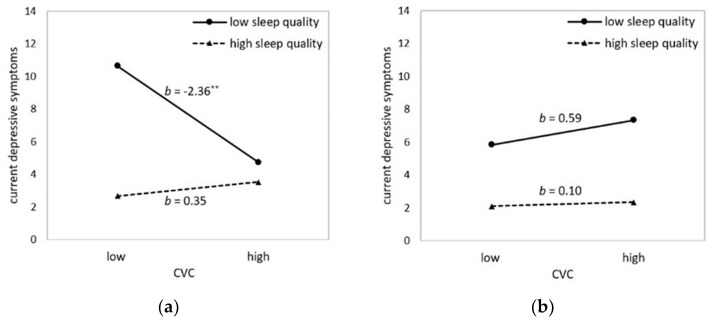
Illustration of the interaction between sleep quality (PSQI) and CVC (HF-HRV) in predicting current depressive symptoms (BDI-II’1): (**a**) in women and (**b**) in men. High sleep quality corresponds to low PSQI scores. Values show estimates at means ±1 SD. Asterisks denote slopes that are significantly different from zero. ** *p* ≤ 0.01.

**Figure 2 ijerph-16-04067-f002:**
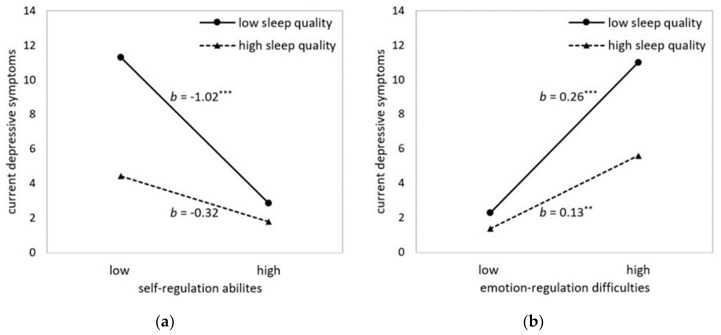
Illustration of the interactions in predicting current depressive symptoms (BDI-II’1): (**a**) between sleep quality and self-regulation abilities (REG) and (**b**) between sleep quality and emotion regulation difficulties (DERS). High sleep quality corresponds to low PSQI scores. Values show estimates at means ±1 SD for all predictors. Asterisks denote slopes that are significantly different from zero. ** *p* ≤ 0.01, *** *p* ≤ 0.001.

**Table 1 ijerph-16-04067-t001:** Descriptives and zero-order correlations.

Variables	M	SD	Range	HF-HRV	REG	DERS	PSQI	BDI-II‘1	BDI-II‘2
**Primary study variables**									
CVC(HF-HRV; ms^2^) ^a^	7.36	1.25		-	-	-	-	-	-
Self-regulation (REG) ^b^	30.86	4.16	10–40	0.18 ^+^	-	-	-	-	-
Emotion regulation (DERS) ^b^	70.62	16.89	36–180	−0.19 *	−0.65 ***	-	-	-	-
Sleep quality (PSQI) ^b^	4.64	2.18	0–21	−0.17 ^+^	−0.28 **	0.36 ***	-	-	-
Depression T1 (BDI-II‘1) ^b^	5.45	5.32	0–57	−0.11	−0.50 ***	0.64 ***	0.57 ***	-	-
Depression T2 (BDI-II‘2) ^c^	5.23	5.77	0–57	0.01	−0.32 **	0.43 ***	0.36 ***	0.62 ***	-
**Control variables**									
Age (years) ^b^	23.58	4.38		−0.05	−0.04	−0.05	−0.14	−0.09	−0.06
Body mass index (BMI, kg/m^2^) ^b^	21.94	3.09		0.15	−0.10	0.04	−0.01	−0.04	−0.04

Descriptive values and Spearman correlations are reported. Larger values indicate better self-regulation (REG), greater difficulty in emotion regulation (DERS), worse sleep quality (PSQI), and more depressive symptoms (BDI-II’1 & 2). Mean values and correlations based on the reduced BDI sum score without items referring to sleep are reported. Mean values of the standard BDI-II score including all items are 6.52 (T1) and 6.25 (T2). ^a^
*n* = 112, ^b^
*n* = 125, ^c^
*n* = 84 (correlation with REG, DERS, PSQI, BDI-II’1) or *n* = 75 (correlation with HF-HRV). ^+^
*p* ≤ 0.10, * *p* ≤ 0.05, ** *p* ≤ 0.01, *** *p* ≤ 0.001.

**Table 2 ijerph-16-04067-t002:** Effects of CVC, subjective sleep quality, and their interaction on depressive symptoms.

Predictors	b	SE (HC3)	t	*p*	95%CI (l; u)
*Cross-sectional* ^a^					
**Model C1**					
CVC (HF-HRV)	−0.51	0.36	−1.41	0.160	−1.22; 0.20
Sleep quality (PSQI)	0.98	0.18	5.53	<0.001 ***	0.63; 1.33
CVC x sleep quality	−0.24	0.15	−1.57	0.120	−0.54; 0.06
**Model C2**					
CVC (HF-HRV)	−1.01	0.49	−2.06	0.042 *	−1.98; −0.04
Sleep quality (PSQI)	1.06	0.23	4.53	<0.001 ***	0.60; 1.52
CVC x sleep quality	−0.62	0.25	−2.55	0.012 *	−1.11; −0.14
Sex	−0.99	0.77	−1.29	0.201	−2.51; 0.53
Sex x CVC	1.35	0.71	1.91	0.059 ^+^	−0.05; 2.76
Sex x PSQI	−0.05	0.41	−0.13	0.899	−0.86; 0.76
Sex x CVC x PSQI	0.74	0.39	1.91	0.059 ^+^	−0.03; 1.50
*Longitudinal* ^b^					
**Model L1**					
Depression T1 (BDI-II‘1)	0.78	0.25	3.10	0.003 **	0.28; 1.28
CVC (HF-HRV)	0.54	0.44	1.23	0.222	−0.34; 1.43
Sleep quality (PSQI)	0.07	0.35	−0.20	0.842	−0.63; 0.77
CVC × sleep quality	0.17	0.26	0.67	0.505	−0.34; 0.68
**Model L2**					
Depression T1 (BDI-II‘1)	0.72	0.25	2.86	0.006 **	0.22; 1.22
CVC (HF-HRV)	−0.07	0.50	−0.14	0.893	−1.05; 0.91
Sleep quality (PSQI)	0.09	0.45	0.19	0.848	−0.81; 0.98
CVC × sleep quality	0.07	0.55	0.12	0.902	−1.03; 1.16
Sex	−0.93	1.61	−0.58	0.567	−4.14; 2.29
Sex × CVC	1.67	1.61	1.04	0.301	−1.53; 4.89
Sex × sleep quality	0.23	1.05	0.22	0.829	−1.86; 2.32
Sex × CVC × sleep quality	0.22	0.94	0.23	0.818	−1.66; 2.10

HF-HRV and PSQI were mean-centered for all analyses and sex was dummy coded (reference category: female). ^a^
*n* = 112, ^b^
*n* = 75. ^+^
*p* ≤ 0.10, * *p* ≤ 0.05, ** *p* ≤ 0.01, *** *p* ≤ 0.001.

**Table 3 ijerph-16-04067-t003:** Effects of self-reported impaired regulatory processes, subjective sleep quality, and their interaction on depressive symptoms.

Predictors	b	SE (HC3)	t	P	95% CI(l; u)
*Cross-sectional ^a^*					
**Model C3**					
Self-regulation (REG)	−0.67	0.13	−4.96	<0.001 ***	−0.93; −0.40
Sleep quality (PSQI)	0.91	0.19	4.83	<0.001 ***	0.53; 1.28
Self-regulation × sleep quality	−0.16	0.06	−2.87	0.005 **	−0.27; −0.05
**Model C4**					
Emotion regulation (DERS)	0.19	0.03	6.27	<0.001 ***	0.13; 0.25
Sleep quality (PSQI)	0.72	0.16	4.53	<0.001 ***	0.41; 1.04
Emotion regulation × sleep quality	0.03	0.01	2.64	0.009 **	0.01; 0.05
*Longitudinal ^b^*					
**Model L3**					
Depression T1 (BDI-II’1)	0.74	0.19	3.87	<0.001 ***	0.36; 1.12
Self-regulation (REG)	−0.04	0.14	−0.31	0.759	−0.33; 0.24
Sleep quality (PSQI)	−0.06	0.33	−0.18	0.855	−0.71; 0.59
Self-regulation × sleep quality	0.06	0.07	0.91	0.364	−0.07; 0.20
**Model L4**					
Depression T1 (BDI-II’1)	0.69	0.20	3.44	<0.001 ***	0.29; 1.09
Emotion regulation (DERS)	0.04	0.04	0.83	0.412	−0.05; 0.13
Sleep quality (PSQI)	−0.03	0.33	−0.10	0.922	−0.68; 0.62
Emotion regulation × sleep quality	−0.02	0.02	−0.95	0.347	−0.05; 0.02

REG, DERS and PSQI were mean-centered for all analyses. ^a^
*n* = 125, ^b^
*n* = 84. ** *p* ≤ 0.01, *** *p* ≤ 0.001.

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
