# Peer review of "Cardiac Vagal Control, Regulatory Processes and Depressive Symptoms: Re-Investigating the Moderating Role of Sleep Quality"

_ijerph, 2019, doi:10.3390/ijerph16214067_

Round 1

Reviewer 1 Report

The authors have developed a thorough study with a very careful design.  It shows the usefulness of studying the validity of assumptions described in the previous literature. The search for simple predictive markers give the research clinical interest.

Besides the limitations the authors pointed out in the last paragraphs of “Conclusion”, I recommend to review the following

-106 Only healthy adults without psychiatric, cardiovascular, neurological or thyroid diseases could participate in the study

The methods or test that have been used for inclusion criteria, to rule out the existence of cardiovascular, neurological or thyroid diseases are missed. As this conditions can modify and alter the interpretation of results, a brief summary of how this pathology has been objectively ruled out would be adequate.

-108 cigarettes a day) or who took medicine at the day of assessment, that might interact

There are two issues that can alter the results, if they have not been taken into account. If investigators have taken them into account, as I suppose, should be reflected in the text.

1-As the paper points out, some drugs modify HRV.

Authors give just some examples of drugs that causes exclusion, because they act on the (para-) sympathetic system. It’s be more accurate to show the list of drugs that they used to decide the no inclusion.

2-The non-use restriction for these drugs just on the assesment day seems insufficient.  There are drugs, such as some antihistamines, with a much longer half-life.

-Regarding the Title

I’m not an expert in depresión diagnosis. In this study, the diagnosis was based in Self Assessment Validated Questionnaires.

I ask the authors: Would it be more accurate the following title?:

Cardiac Vagal Control, Regulatory Processes and Self Assessed Depressive Symptoms: Re-investigating the Moderating Role of Sleep Quality

Author Response

Reviewer #1

The authors have developed a thorough study with a very careful design.  It shows the usefulness of studying the validity of assumptions described in the previous literature. The search for simple predictive markers give the research clinical interest. Besides the limitations the authors pointed out in the last paragraphs of “Conclusion”, I recommend to review the following.

Comment #1: 106 Only healthy adults without psychiatric, cardiovascular, neurological or thyroid diseases could participate in the study

The methods or test that have been used for inclusion criteria, to rule out the existence of cardiovascular, neurological or thyroid diseases are missed. As this conditions can modify and alter the interpretation of results, a brief summary of how this pathology has been objectively ruled out would be adequate.

Response: As assessing psychiatric, cardiovascular, neurological and thyroid diseases objectively was beyond the scope of this study, we had to rely on participants’ self-report. Therefore, we used a self-developed questionnaire asking whether participants ever had one of those diseases and if yes which one(s). We included this information in our manuscript as well (l. 106, “Only healthy adults without self-reported psychiatric, cardiovascular, neurological or thyroid diseases could participate in the study (overall inclusion criteria).”).

Comment #2: 108 cigarettes a day) or who took medicine at the day of assessment, that might interact

There are two issues that can alter the results, if they have not been taken into account. If investigators have taken them into account, as I suppose, should be reflected in the text.

1-As the paper points out, some drugs modify HRV. Authors give just some examples of drugs that causes exclusion, because they act on the (para-) sympathetic system. It’s be more accurate to show the list of drugs that they used to decide the no inclusion.

2-The non-use restriction for these drugs just on the assesment day seems insufficient.  There are drugs, such as some antihistamines, with a much longer half-life.

Response: Thank you for pointing out these issues.

To address your first concern, we now provide a complete list of all drugs which caused exclusion from the HF-HRV analyses (ll. 109-110, “antihistamines, pain killers, sympathomimetics, cortisone and sumatriptan”) in the revised manuscript.

Please note that this list was derived post-hoc based on the actual drug-use participants reported, because it would have been unfeasible to provide a full list of all drugs which could affect HF-HRV a-priori. Of all drugs reported by participants, only the mentioned drugs (antihistamines, pain killers, sympathomimetics, cortison and sumatriptan) were considered to possibly affect HF-HRV which led to exclusion of participants who had taken these drugs on the assessment day from HF-HRV analyses. The use of hormonal contraceptives, urovaxom and omeprazole was also reported by some participants. However, as these drugs are unlikely to influence HF-HRV, we did not exclude participants who had taken these drugs.

Regarding your second concern, we carefully re-considered the temporal restrictions for non-use of the critical drugs based on their half-life. The only HF-HRV-relevant drugs which have been taken by participants of our current HF-HRV sample during the 4 days before the assessment day were pain killers and antihistamines. As pain killers usually have short half-times, we decided to stick to only excluding participants who have taken such on the assessment day. For antihistamines, half-times vary greatly between different compounds. Therefore, we decided to look at the specific compounds reported by participants of our current HF-HRV sample and their half-times and decide case-wise (2 cases of antihistamine use during the 4 days before the assessment). One participant took Levocetirizin (half time: 7.9+/-1.9h) in the evening and was assessed at 12:41 o’clock the following day. The other participant took Cetirizin (half time: 10h) at morning and was assessed at 10:07 o’clock on the following day. Considering these half-times in combination with the time period between the drug intake and the assessment, we concluded that an influence on HF-HRV was very unlikely. Thus, we decided to refrain from additionally excluding these participants.

Comment #3: Regarding the Title

I’m not an expert in depresión diagnosis. In this study, the diagnosis was based in Self Assessment Validated Questionnaires. I ask the authors: Would it be more accurate the following title?: Cardiac Vagal Control, Regulatory Processes and Self Assessed Depressive Symptoms: Re-investigating the Moderating Role of Sleep Quality

Response: Thank you for your interesting question. First, the current assessment of depressive symptoms cannot be interpreted as diagnosis (which we never claimed throughout the manuscript). Though self-reported depressive symptoms above a certain cut-off can indicate the presence of clinically relevant depression, we refrain from speaking about depression (diagnosis) throughout the manuscript as we are more interested in the dimensional aspect of depressive symptoms.

Second, as depressive symptoms and depression are always assessed via self-report (in both, questionnaires and interviews), the expression “self-assessed depressive symptoms” would be redundant. Therefore, we decided to refrain from adapting the title.

Reviewer 2 Report

Review is attached in a Word document.

Author Response

Reviewer #2

This is an interesting manuscript and the authors have done a lot of work. I think that with some changes the paper could be a good fit for IJERPH, but it needs some revision first, as I have some concerns, which are detailed below.

Comment #1: Throughout the article there are grammatical errors that need to be addressed by a native English proofreader – for example, in the abstract there are already a number of errors, such as: ‘…often understood as indicator should be ‘an indicator’. There are lots of errors like this throughout which are not the job of a scientific reviewer to correct so I’d suggest a proofreader before the manuscript is published.

Response: Thank you for pointing that out. Due to time issues the editor now decided to publish the paper without the proof reading, which would have needed another 2 weeks time. / We now had the paper grammar checked by a native English proofreader.

Comment #2: Of the participants who dropped out at follow-up, did the authors check that this was not thus introducing some form of systematic bias to their results for the follow-up? It is quite common to check whether those who dropped out differ from those who remained in the study, perhaps with some kind of attrition analysis to show the reader that they don’t have a biased sample at follow-up. Also, importantly, the authors gave those who came back at 3 months the chance to enter into a lottery for a voucher, so this may introduce some bias too.

Response: Thank you for this useful idea. We now did so by conducting t-tests to explore differences in all primary variables (HF-HRV, REG, DERS, PSQI, BDI-II’1) between those who stayed in our sample and those who dropped out before our follow-up assessment. No differences in all variables emerged (ps ≥ .613) indicating no systematic bias for the follow-up. This check-up was included as a footnote in the revised manuscript (ll. 122 “[1] Participants who dropped out did not differ from completers in CVC, self-regulation, emotion regulation, sleep quality and depressive symptoms, ps ≥ .613”).

Comment #3: I think there needs to be much better and solid justification of why only BMI and age were used as covariates in the models – surely there are also other factors that might be important to consider here? Please provide a more detailed explanation for the choice of covariates in the Methods section.

Response: Thank you for your comment. We now added more detailed information for the choice of covariates in the methods section, which was based on the guidelines for reporting articles on psychiatry and heart rate variability (ll. 210-211 “Additionally, on the basis of current recommendations [39], links with further confounding variables (i.e., age, BMI) were examined.”). According to Quintana, Alvares & Heathers (2016) the most important variables include psychiatric disorders, gender, age, physical activity levels, alcohol and nicotine intake as well as medication. In our study, we first ruled out psychiatric as well as additionally cardiovascular and neurological disorders as part of the exclusion criteria. Second, gender was taken into account by including this variable in our main analyses for exploratory purposes. Third, in our self-developed questionnaire we asked for alcohol, nicotine and medication use. Only nicotine and medication were indicated by some participants and, as described in the manuscript and this letter, we further excluded participants who smoked on a regular basis or took medicine that might influence heart rate variability. Finally, we controlled for BMI – as index for physical activity levels (see Bovet, Auguste, & Burdette, 2007; Nikolaidis, 2013) – as well as for age. Therefore, all recommended confounding variables were taken into account.

Comment #4: Can the authors comment on any potential issues with performing 6 t-tests to check for sex differences in their primary variables? I should think that ANOVAs are better because they are more robust and control type-1 error rates, which are a problem with t-tests.

Response: As far as we know, ANOVAS and t-tests do not differ in robustness and type-1 error rate. If you refer to a MANOVA, this analysis would not be appropriate for our data as it is usually thought for closely related dependent variables.

Furthermore, given that these t-tests to check for sex differences are only conducted for control purposes, we do not think that alpha inflation would be problematic as this would just make our control analyses for biases more sensitive.

Comment #5: I’m not sure whether the authors should call the results for the 3-month follow-up longitudinal, as such, but perhaps just call those results ‘follow-up’ or something similar?

Response: We understand your concerns, as the length of our follow-up interval is one of the major limitations of our study. However, with using the term longitudinal we did not want to over-exaggerate the meaning of our follow-up but wanted to stress the difference to the study we were replicating which does not have a follow-up at all and is merely cross-sectional. To stress this extension of the replication we decided to keep calling these results longitudinal. As already described, we also clearly mention the length of our follow-up interval as limitation. However, if the reviewer and editor continue to think that this term is misleading we would of course change that in another revision.

Comment #6: I think the main results tables for regression models don’t really need to include: the t value; also the column for the confidence intervals should probably be labelled ‘95%CI’, as I don’t think the way it’s labelled at the moment is common. I’d say with reporting results in the text, also remove the t-value.

Response: Thanks for pointing this out, we re-labelled the columns for the confidence intervals. However, to keep comparability with previous studies in this research field, we would like to refrain from removing the t-values but could of course still do that in another revision.

Comment #7: Is there any evidence that for example, the relationship might be sort of bidirectional, such that there might be interactions between CVC and depressive symptoms on sleep quality (as an outcome)? This might be worth discussing as a possibility, as many of these incredibly complex relationships are often bidirectional, but of course this isn’t possible to assess without proper prospective data over many years or some form of proper causal analysis.

Response: Thank you for that interesting comment. As far as we know, there are no studies looking at the influence of an interaction between CVC and depressive symptoms on sleep quality. However, we agree with the reviewer that these relationships are often bidirectional and there are studies indicating bidirectional links between depressive symptoms and sleep. For example, it has been shown that in cases of comorbid depression and insomnia, insomnia occurs first in more than 40%, whereas in around 38% depression occurs first (Franzen & Buysse, 2008). This indicates that there are cases where insomnia precedes depression and other cases where depression exceeds insomnia. Overall, it seems – however – that sleep is a more important predictor for depression given that it strongly increases the risk for the onset or a new depressive episode as well as for other mental disorders (Baglioni et al., 2011; Hertenstein et al., 2019). Although there is the possibility that an interaction between CVC and depressive symptoms influences sleep quality, we would prefer to not open this discussion in the current paper because we feel that an adequate and satisfactorily discussion of this topic is beyond the scope of this paper.

Comment #8: In the Discussion section the authors imply that if they had longitudinal data >1 year on their participants they may be able to ascertain temporal associations, which is only true to some extent, as

Response: Unfortunately, a part of this comment is lacking. If necessary, we are of course happy to address this concern in another revision.

Comment #9: The authors should refrain from using the word ‘significant’ as often as they do, as this is now becoming less commonplace, with all of the discussions around p-values and significance thresholds. They can say exactly what they want to say and draw the same conclusions if they remove this word throughout the manuscript.

Response: Thank you for this advice. We now reduced our usage of the term “significant”, especially in the discussion section.

Comment #10: An important point is also the fact that in their limitations, the authors do not acknowledge some crucial things such as:

  1. BMI is calculated from self-reported height and weight, which has major issues because people tend to overestimate their height and underestimate their weight, so some discussion of this is definitely necessary and please include references. Objective height and weight measured by a nurse or similar would be much better, but I understand if this is not available in the study.
  2. There is only mention of how large and representative their sample size is in the Introduction, but an N of 125 is not that large and particularly as they then lost several people at the 3-month follow-up. Some discussion of this is necessary in the limitations section. Also, regarding representativeness, the authors just tell us that their sample is representative, but provide no references or data to back this up.
  3. Why was only the reassessment of depressive symptoms used for follow-up analyses, when other measures were also taken? This is fine if the authors have a justification that they can include.
  4. Also, the authors need to acknowledge potential issues with the fact that they collected measures at baseline and follow-up differently – they said that the follow-up questionnaires were completed online. Could this introduce any issues or biases?

Response: Thank you for this detailed feedback on the limitations of our study.

  1. To address your concern, we included the self-reported height and weight and the associated potential bias as a limitation of the current study (ll.393-394, “Fourth, the calculation of the BMI was based on self-reported height and weight which might differ from objective assessments [45]”).
  2. Following your recommendation, we mentioned the drop-out at follow-up as another limitation of the current study and discussed it appropriately (ll.383-386, “Second, some participants dropped out before follow-up which reduced the power for our longitudinal analyses. However, as our a priori sample-size calculation indicated a need for at least 66 participants, we feel confident that our longitudinal analyses (based on at least 75 participants) were not underpowered.”).

The representativeness of our sample can be evaluated by looking at the beginning of the results section (ll.236-241): The mean BMI and age in our sample were rather low and more women than men could be recruited. However, the percentage of people suffering from clinically relevant depressive symptoms (between 8 and 11% of the current sample) and from sleep disturbances (30% of the current sample) was quite representative.

  1. We only used this variable for follow-up analyses, as this was our only longitudinal hypothesis derived from empirical evidence showing influences of CVC, regulatory processes and sleep quality on subsequent depressive symptoms.
  2. We carefully considered your concern and concluded that the way of assessment (in the lab vs. at home) was unlikely to bias the responses of the participants as both assessments were 1) computer-based 2) identically designed and 3) anonymous.

Comment #11: I don’t think that there is sufficient justification in the Introduction for why the authors would want to place so much emphasis on investigating sex differences? There should be a better rationale for this.

 Response: We agree with the reviewer that there are only preliminary indications that sex differences might play a role in the relationship between CVC, sleep quality and depressive symptoms. Based on that, we decided to include sex as additional, exploratory variable in the main analyses instead of controlling for it like we did for age and BMI like described in the manuscript (ll. 91-93).

Comment #12: The authors seem to imply, in the Introduction, that previous studies have been unable to determine causality due to the use of cross-sectional data. This is true, but their study is also completely unable to do this, as (also acknowledged by the authors) their follow-up is too short and they do not use any specific causal modelling either.

 Response: We would like to point out that this is the first study investigating the influence of the interaction of CVC and sleep quality on later depressive symptoms. We agree with the reviewer that, due to the non-significant findings with regard to the follow-up assessment, we were not able to determine a causal effect on depressive symptoms. While this may be due to the short follow-up interval, we argue that these are still valuable preliminary results that need further investigation. As mentioned, we included this point in the discussion and limitations of the manuscript.

Comment #13: I think that perhaps some sensitivity analyses of those participants who were in both the baseline and follow-up (as a separate group) might be appropriate, as this gives the authors a complete case dataset to draw conclusions from. At the moment, they are drawing conclusions from two different groups of individuals, which can make it more difficult to interpret, as they are not only examining change in the same participants. I would suggest some additional analysis in only the n=75 who were included at both baseline and follow-up to see whether the results are the same or not as what they have found using the two different groups.

Response: We carefully considered your concern and decided to refrain from conducting any further analyses with the group of participants which took part in both, baseline and follow-up assessment. In our opinion, such an approach would have no additional value for our paper: All reported longitudinal analyses are already based on this subgroup of participants which took part in both assessments (this is the standard procedure in longitudinal analyses which allows to examine within-person changes), and thus your concern does not apply to our longitudinal analyses. For all reported cross-sectional analyses, an artificial reduction of the sample size would have no additional value (as we have already shown that no systematic bias is causing dropout rates, see comment #2) but would cause an unnecessary reduction of power. Moreover, ethical considerations also speak against picking out only a subset of all available data for analyses.

Comment #14: Are there any issues that need to be considered when analysing a sample that is predominantly females and then drawing conclusions from analyses of sex differences? Usually for parametric analyses like t-tests (or ANOVAs) one needs to have (as much as possible) even numbers in each group.

Response: For the independent t-tests we used to explore sex differences in primary variables (to explore whether our data was biased), even numbers in each group do not constitute an assumption (Bühner & Ziegler, 2017). According to Bühner and Ziegler (2017), independent t-tests only require interval scaled measures, no dependencies between subjects (i.e., no nested designs possible), normal distribution of data within each group and homogeneity of variances. Prior to all analyses, these assumptions were checked and taken into account following Bühner and Ziegler (2017).

Comment #15: Are the authors’ findings generalisable to more middle-aged/older adults? The age range isn’t particularly wide so this might be important to acknowledge. Perhaps we still need studies of middle-aged and older adults, especially as those in middle age are more likely to suffer from some of the problems that the authors are investigating here.

Response: Thank you for this question. In general, there are no reasons why the findings should differ in middle-aged or older adults. As you point out, middle-aged and older individuals are even more likely to suffer from sleep-related problems and depressive symptoms. Therefore, empirical studies examining these relationships would be absolutely relevant and might lead to new targets for psychological interventions in these populations.

Comment #16: Does the PROCESS program by Andrew Hayes correctly handle interactions between variables that are not continuous?

Response: Yes, accordingly to the author it does. For further information please see Hayes (2017).

Comment #17: Might it be worth discussing what the issues are with self-reported sleep measures? This is quite a big thing and it is well known that we often don’t necessarily find the same results with objective sleep measures.

Response: Thank you very much for your interest in this topic. As you point out, self-reported versus objectively measured sleep is an often discussed topic and results can greatly very depending on the used measures. However, there are several reasons why we would like to keep the focus on subjective sleep quality and think that discussing this topic is beyond the scope of this paper. Most importantly, the subjective “sleep quality” experience seems to cover additional aspects that cannot be exhaustively captured via objective indices yet (see Krystal & Edinger, 2008 for a discussion on this topic). In line with this, it has been shown that the phenomenon of paradoxical insomnia (i.e., reporting subjective complaints while having comparable objective sleep indices to individuals without subjective complaints) is quite common (Carskadon et al., 1976; Krystal, Edinger, Wohlgemuth, & Marsh, 2002). Nevertheless, the diagnosis of insomnia disorder is only based on subjective complaints (e.g., Harvey & Spielman, 2011). Therefore, we would like to refrain from discussing these issues in the current manuscript.

Comment #18: There’s no Cronbach’s alphas reported for the PSQI – is there a reason for this?

Response: Thank you for noting this mistake. We now added Cronbach’s alpha for the PSQI (ll. 195-196).

Comment #19: On page 6, where the authors mention that there were no sex differences, except that CVC was marginally higher in women than in men, as that’s not a real (‘significant’) difference at the 0.05 alpha level, so unless you’re using 0.1 then I’d remove that because it’s not great to make a big deal out of this

Response: Thank you for your comment. In order to be more precise, we would prefer to stick to reporting this although we of course refrain from interpreting this result more closely. However, but if the reviewer and editor continue to think that this is not necessary, we are of course willing to remove it.

In conclusion, we are grateful to the reviewers for their helpful comments which guided our revision and further improved our manuscript. We hope that it is now acceptable for publication in the IJERPH.

Cited Literature (in the manuscript)

  1. Quintana, D.S.; Alvares, G.A.; Heathers, J.A.J. Guidelines for Reporting Articles on Psychiatry and Heart rate variability (GRAPH): recommendations to advance research communication. Transl. Psychiatry 2016, 6, e803–e803.
  2. Gorber, S.C.; Tremblay, M.; Moher, D.; Gorber, B. A comparison of direct vs. self-report measures for assessing height, weight and body mass index: a systematic review. Obes. Rev. 2007, 8, 307–326.

Additional cited literature (in this response letter)

Baglioni, C., Battagliese, G., Feige, B., Spiegelhalder, K., Nissen, C., Voderholzer, U., . . .

Riemann, D. (2011). Insomnia as a predictor of depression: a meta-analytic evaluation of longitudinal epidemiological studies. Journal of Affective Disorders, 135(1-3), 10-19. doi:10.1016/j.jad.2011.01.011

Bovet, P., Auguste, R., & Burdette, H. (2007). Strong inverse association between physical fitness and overweight in adolescents: a large school-based survey. International Journal of Behavioral Nutrition and Physical Activity, 4, 24. doi:10.1186/1479-5868-4-24

Bühner, M., & Ziegler, M. (2017). Statistik für Psychologen und Sozialwissenschaftler (2nd

            ed.). Hallbergmoos, Deutschland: Pearson.

Carskadon, M. A., Dement, W. C., Mitler, M. M., Guilleminault, C., Zarcone, V. P., &

Spiegel, R. (1976). Self-reports versus sleep laboratory findings in 122 drug-free subjects with complaints of chronic insomnia. The American Journal of Psychiatry, 133(12), 1382-1388.

Franzen, P. L., & Buysse, D. J. (2008). Sleep disturbances and depression: risk relationships for subsequent depression and therapeutic implications. Dialogues in Clinical Neuroscience, 10(4), 473–481.

Harvey, A. G., & Spielman, A. J. (2011). Insomnia: Diagnosis, Assessment and Outcomes. In M. H. Kryger, T. Roth, & W. C. Dement (Eds.), Principles and Practice of Sleep Medicine (5th ed., pp. 838-849): Philadelphia: Elsevier Saunders.

Hayes, A. F. (2017). Introduction to mediation, moderation, and conditional process analysis:

A regression-based approach (2nd ed.). New York, NY: Guilford Publications.

Hertenstein, E., Feige, B., Gmeiner, T., Kienzler, C., Spiegelhalder, K., Johann, A., ...

Baglioni, C. (2019). Insomnia as a predictor of mental disorders: A systematic review and meta-analysis. Sleep Medicine Reviews, 43, 96-105. doi:10.1016/j.smrv.2018.10.006

Krystal, A. D., & Edinger, J. D. (2008). Measuring sleep quality. Sleep Medicine, 9 Suppl 1, S10-17. doi:10.1016/s1389-9457(08)70011-x

Krystal, A. D., Edinger, J. D., Wohlgemuth, W. K., & Marsh, G. R. (2002). NREM sleep EEG frequency spectral correlates of sleep complaints in primary insomnia subtypes. Sleep, 25(6), 630-640.

Nikolaidis, P. T. (2013). Body mass index and body fat percentage are associated with decreased physical fitness in adolescent and adult female volleyball players. Journal of Research in Medical Sciences, 18(1), 22-26.